# Functional Analysis of Pheromone Biosynthesis Activating Neuropeptide Receptor Isoforms in *Maruca vitrata*

**DOI:** 10.3390/cells12101410

**Published:** 2023-05-17

**Authors:** Wook Hyun Cha, Boyun Kim, Dae-Weon Lee

**Affiliations:** 1Department of SmartBio, Kyungsung University, Busan 48434, Republic of Korea; whcha17@gmail.com (W.H.C.);; 2Metabolomics Research Center for Functional Materials, Kyungsung University, Busan 48434, Republic of Korea

**Keywords:** sex pheromone, PBAN receptor, confocal microscopy, RNA interference, gas chromatography, *Maruca vitrata*

## Abstract

Insect sex pheromones are volatile chemicals that induce mating behavior between conspecific individuals. In moths, sex pheromone biosynthesis is initiated when pheromone biosynthesis-activating neuropeptide (PBAN) synthesized in the suboesophageal ganglion binds to its receptor on the epithelial cell membrane of the pheromone gland. To investigate the function of PBAN receptor (PBANR), we identified two PBANR isoforms, MviPBANR-B and MviPBANR-C, in the pheromone glands of *Maruca vitrata*. These two genes belong to G protein-coupled receptors (GPCRs) and have differences in the C-terminus but share a 7-transmembrane region and GPCR family 1 signature. These isoforms were expressed in all developmental stages and adult tissues. MviPBANR-C had the highest expression level in pheromone glands among the examined tissues. Through in vitro heterologous expression in HeLa cell lines, only MviPBANR-C-transfected cells responded to MviPBAN (≥5 µM MviPBAN), inducing Ca^2+^ influx. Sex pheromone production and mating behavior were investigated using gas chromatography and a bioassay after MviPBANR-C suppression by RNA interference, which resulted in the major sex pheromone component, E10E12-16:Ald, being quantitatively reduced compared to the control, thereby decreasing the mating rate. Our findings indicate that MviPBANR-C is involved in the signal transduction of sex pheromone biosynthesis in *M. vitrata* and that the C-terminal tail plays an important role in its function.

## 1. Introduction

In moths, the females’ ability to attract conspecific males is important for successful reproduction. Female moths secrete species-specific volatile compounds known as sex pheromones, which induce mating behavior [1,2]. Sex pheromones are biosynthesized in the pheromone glands located on the 8th and 9th abdominal segments of female moths. Most moths biosynthesize and secrete Type I sex pheromones from acetyl-CoA, an intermediate product of fatty acid biosynthesis, through various biochemical processes such as desaturation and functional group modification [3,4,5]. The biosynthesis and secretion of sex pheromones are regulated by the diurnal rhythm, and production is mainly increased in the scotophase [6].

Sex pheromone biosynthesis in moths is triggered by pheromone biosynthesis activating neuropeptide (PBAN) that is synthesized in the subesophageal ganglion (SEG) [7]. PBAN is a neuropeptide that is 33–34 aa in length and belongs to the pyrokinin/PBAN family. Most lepidopteran PBANs share FSPRL-NH_2_ in the C-terminal, which is essential for their activity [8]. Signal transduction of sex pheromone biosynthesis by PBAN is initiated by PBAN binding to the PBAN receptor (PBANR) located in the epidermal cells of the pheromone glands [9]. Subsequently, the levels of secondary messengers such as intracellular Ca^2+^ and cyclic AMP (cAMP) are elevated [10,11].

PBANR belongs to the G protein-coupled receptor (GPCR) family that includes a 7-transmembrane (TM) domain and is an ortholog to the mammalian neuromedin U receptor [12]. Since PBANR was first identified in *Helicoverpa zea*, it has been found in several other moth species, including *Bombyx mori*, *Spodoptera exigua*, *Plutella xylostella*, and *Antheraea pernyi* have been reported so far [13,14,15,16,17]. PBANR is divided into three isoforms (PBANR-A, -B, and -C) according to the length and composition of the C-terminus. PBANR-A contains a short and AT-rich C-terminal, whereas PBANR-B and -C contain a GC-rich and relatively long C-terminal [18,19]. These differences in PBANR isoforms are caused by alternative splicing and affect ligand-induced internalization [20].

*Maruca vitrata* is an agricultural pest that causes serious damage to leguminous crops such as soy and mung beans, and is distributed throughout Asia, Australia, and Africa [21,22]. Sex pheromones of *M. vitrata* consist of one main component, (E,E)-10,12-hexadecadienal (E10E12-16:Ald), and two minor components, (E,E)-10,12-hexadecadienol and (E)-10-hexadecenal [23]. In previous studies, the sex pheromone biosynthesis pathway of *M. vitrata* was predicted based on the transcriptome analysis of the pheromone glands [24]. In addition, PBAN, which induces sex pheromone biosynthesis, was found in the heads of female adults of *M. vitrata* [25]. Functional analysis using RNA interference (RNAi) revealed that pheromone gland-specific fatty acyl reductase (pgFAR) was also involved in the sex pheromone biosynthesis of *M. vitrata* [26]. However, signal transduction studies on PBANR in *M. vitrata* remain insufficient. Therefore, in this study, PBANR isoforms presumed to be involved in signal transduction of sex pheromone biosynthesis were identified in the pheromone glands of *M. vitrata*, and the function of signal transduction was investigated by measuring Ca^2+^ influx via in vitro heterologous expression. After the suppression of PBANR isoform function in signal transduction by RNAi, quantitative change in major sex pheromone components and mating rates were analyzed.

## 2. Materials and Methods

### 2.1. Insects

*M. vitrata* larvae were reared on an artificial diet under the following conditions: temperature, 25 ± 1 °C; relative humidity, 60%; and a 16:8 h light-dark cycle (L:D) (light period of 3 a.m. to 7 p.m.) [27]. Female pupae were separated prior to eclosion in an insect breeding dish to obtain virgin adult females. Adults were reared on a diet of 10% sucrose in an acrylic breeding box (50 × 50 × 50 cm; Gaia Co., Republic of Korea).

### 2.2. Cloning of PBANR Isoforms

PBANR isoforms (MviPBANR-B and -C) were cloned from the pheromone glands of *M. vitrata.* Pheromone glands were dissected from 20 two-day-old virgin adult females. Total RNA was extracted from dissected pheromone glands using Trizol^®®^ reagent (Ambion, Austin, TX, USA), according to the manufacturer’s instructions. Complementary DNA (cDNA) was synthesized from 1 μg of total RNA using the TOPscript™ cDNA Synthesis Kit (Enzynomics, Republic of Korea), according to the manufacturer’s instructions. Then synthesized cDNA was incubated at 37 °C for 20 min to remove RNA by adding RNase H (Enzynomics, Republic of Korea).

Before MviPBANR-B and -C were amplified, gene-specific primers were designed and synthesized based on the sequence of our previous transcriptome data (Appendix A) [24]. MviPBANR isoforms were amplified from the cDNA of pheromone glands by PCR with 2× TOPsimple™ DyeMIX-HOT (Enzynomics, Republic of Korea) and specific primers (Appendix A). The following parameters were used to amplify MviPBANR isoforms from cDNA: 95 °C for 10 min, followed by 40 cycles at 95 °C for 30 s, 63 °C for 30 s, and 72 °C for 2 min, and an additional cycle at 72 °C for 5 min. The amplified DNA products were electrophoresed and purified using an EZ-Pure™ Gel Extraction Kit (version 2; Enzynomics, Republic of Korea). The purified products were cloned into a pGEM^®^-T Easy Vector (Promega, Madison, WI, USA) and sequenced (Macrogen Co., Republic of Korea).

### 2.3. Phylogenetic Analysis of Lepidopteran PBANR Isoforms

The deduced amino acid sequences of MviPBANR-B and -C were compared to those of other lepidopteran PBANRs. The protein sequences were aligned with ClustalW using MegAlign software (version 7.1.0). A phylogenetic tree was constructed with maximum likelihood, and bootstrap analysis with 1000 replications was performed using Molecular Evolutionary Genetic Analysis software (version X) [28].

### 2.4. Developmental Stage and Tissue-Specific Expression of PBANR Isoforms

To examine the expression pattern of MviPBANR-B and -C in each developmental stage (eggs, 1st–5th instar larvae, pupae, and adults) and adult tissues (head, thorax, abdomen without pheromone gland or hair pencil, pheromone glands, and hair pencils), total RNA extraction and cDNA synthesis were performed. Quantitative real-time PCR (qRT-PCR) was performed with TOPreal™ qPCR 2× PreMIX (SYBR Green with low ROX) (Enzynomics, Republic of Korea) and specific primers (Appendix A). The following parameters were used for the qRT-PCR: 95 °C for 10 min, followed by 50 cycles at 95 °C for 30 s, 60 °C for 30 s and 72 °C for 30 s. Experiments were repeated independently in triplicate and elongation factor 1-α(EF1-α) and ribosomal protein S18 (RPS18) were used as positive controls. The relative expression levels of MviPBANR-B and -C were calculated using the 2^−ΔΔCT^ method [29].

### 2.5. Transformation for In Vitro Heterologous Expression of MviPBANR Isoforms

Prior to transformation for heterologous expression, the full lengths of MviPBANR-B and -C sequences were amplified from the pheromone glands of *M. vitrata*. Total RNA extraction and cDNA synthesis were performed in the same manner as described above. A PCR was performed with 2× TOPsimple™ DyeMIX-HOT (Enzynomics, Republic of Korea) and specific primers containing *Kpn*I and *Hind*III sequences (Appendix A). The following parameters were used for the PCR: 95 °C for 10 min, followed by 40 cycles at 95 °C for 30 s, 63 °C for 30 s, and 72 °C for 2 min, and an additional cycle at 72 °C for 5 min. The amplified PCR products were electrophoresed and purified using an EZ-Pure™ Gel Extraction Kit (version 2; Enzynomics, Republic of Korea). The purified products were cloned into a pcDNA™3.1/Zeo vector (Invitrogen, Waltham, MA, USA), resulting in expression vectors, pcMviPBANR-B and pcMviPBANR-C.

### 2.6. Confocal Microscopy for Calcium Assay

The expression vectors, pcMviPBANR-B and -C, were transfected into human cervical cancer (HeLa) cell lines using lipofectamine 2000 (Thermo Fisher Scientific, Waltham, MA, USA). The HeLa cell lines were purchased from Korean Cell Line Bank (Seoul, Republic of Korea), and were cultured in RPMI-1640 (Welgene, Republic of Korea) containing 4.5 g/L D-glucose, 2 mM L-glutamine, 10 mM HEPES, 1 mM sodium pyruvate, 1.5 g/L sodium bicarbonate, 10% FBS (Gibco, Grand Island, NY, USA), 100 U/mL penicillin, and 100 g/mL streptomycin (Gibco) The cultured cells were maintained at 37 °C in a humidified atmosphere of 5% CO_2_. The transfected cells with MviPBANR isoforms were selected with 100 µg of zeocin following transfection and maintained for 2 weeks. The transfected HeLa cells (3 × 10^4^/cm^2^) were cultured on a flux-bottom confocal dish for 24 h using a Fluo-4 direct^TM^ calcium assay kit (Thermo Fisher Scientific) according to the manufacturer’s instructions. Briefly, the transfected cells were incubated with Fluo-4 direct calcium assay reagent solution diluted in culture medium (1:1) at 37 °C for 40 min. The alteration of calcium localization following injections of various concentrations of PBAN or 2 μM ionomycin (positive control) was detected every second for 1 min. Time-lapse imaging of calcium signals was taken using a Nikon AX confocal microscope (Nikon, Japan).

### 2.7. Measurement of the Diurnal Expression Level of MviPBANR-C

To examine diurnal expression levels of MviPBANR-C in the pheromone glands of *M. vitrata*, total RNA extraction, and cDNA synthesis were performed every 4 h, starting 2 h before the photophase (1 a.m.), as described above. qRT-PCR was performed with TOPreal™ qPCR 2× PreMIX (SYBR Green with low ROX) (Enzynomics, Republic of Korea) and specific primers (Appendix A). The following parameters were used for qRT-PCR: 95 °C for 10 min, followed by 50 cycles at 95 °C for 30 s, 60 °C for 30 s and 72 °C for 30 s. The experiments were repeated independently in triplicate, and EF1-α and RPS18 were used as positive controls. The relative expression levels of MviPBANR-C were calculated using the 2^−ΔΔCT^ method [29].

### 2.8. Suppression of MviPBANR-C by RNAi

RNAi was used to suppress MviPBANR-C expression in *M. vitrata*. MviPBANR-C-specific double-strand RNAs (dsRNAs) fragments were amplified with a specific primer set containing a T7 promoter region at both ends (Appendix A). MviPBANR-C-specific dsRNAs were directly synthesized using MEGAscript RNAi Kit (Ambion), according to the manufacturer’s instructions. Briefly, dsRNAs were synthesized at 37 °C for 5 h and inactivated at 75 °C for 5 min. The synthesized dsRNAs were mixed with Metafectene^®^ PRO (Biontex, Germany) in a 1:1 ratio and incubated at 25 °C for 30 min. dsRNAs (200 ng) were injected into the female pupae 1 d before eclosion. Enhanced green fluorescent protein (EGFP)-specific dsRNAs were also injected as a negative control. EGFP-specific dsRNAs containing the T7 promoter region at both ends were amplified using specific primers (Appendix A) [30] as described above. To confirm MviPBANR-C suppression in the pheromone glands, the expression level of MviPBANR-C was measured using qRT-PCR for 4 d after dsRNA injection as described above.

### 2.9. Gas Chromatography (GC) Analysis

To investigate the function of MviPBANR-C on sex pheromone biosynthesis in *M. vitrata*, GC analysis was performed to quantify the major sex pheromone component, E10E12-16:Ald, after the suppression of MviPBANR-C via RNAi. Before sex pheromones were extracted from the pheromone glands for GC analysis, 15 two-day-old virgin females were injected with 1 µM of MviPBAN (5′-IPDALPVTPSDDDVYSFKPDSGEVDRRTSYFDPRL-NH_2_-3′) [25] and placed in the dark for 2 h. Subsequently, the pheromone glands were dissected and soaked in hexane for 20 min to extract sex pheromone components. Butylated hydroxytoluene (2,6-di-tert-butyl-4-methylphenol, BHT; Sigma, St. Louis, MO, USA), as an internal control, was added to the extracted sex pheromones and concentrated with nitrogen gas, followed by GC analysis. The standard curve for pheromone quantification was obtained from different concentrations of BHT. Two dimensional GC Model 7890 FID facility (LECO, St. Joseph, MI, USA; Metabolomics Research Center for Functional Materials, Kyungsung University) equipped with a Rxi-5-ms column (30 m × 0.25 mm × 0.25 µm; Bellefonte, PA, USA) was used in this experiment. The following parameters were used: 60 °C for 2 min, increased at a rate of 5 °C/min to 180 °C, maintained for 1 min, then increased at a rate of 10 °C/min to 250 °C, and maintained for 7 min. The experiment was repeated in triplicate.

### 2.10. Mating Rates after Suppression of MviPBANR-C by RNAi

To examine the effect of MviPBANR-C on mating behavior, suppression of PBANR-C by RNAi was performed as described above. The mating behavior of 10 adult male and female pairs was examined, one pair at a time, independently in an acrylic breeding box (50 × 50 × 50 cm; Gaia Co., Republic of Korea) for 3 d. The mating behavior was determined by the presence of the fertilized eggs. The mating rate was defined as the ratio of fertilizing egg-laying pairs to the total examined mating ones. Experiments were performed independently and repeated in triplicate.

### 2.11. Statistical Analysis

ANOVA and *t*-tests were used to analyze statistically significant differences using the Sigmaplot software (version 10.0; Systat software Inc., San Jose, CA, USA).

## 3. Results

### 3.1. Cloning of PBANR Isoforms from Pheromone Gland of M. vitrata

In our previous study, we identified two PBANR isoforms, MviPBANR-B and -C, from the transcriptome of the pheromone glands of *M. vitrata* [24]. In the present study, the full-length of MviPBANR-B and MviPBANR-C were cloned. MviPBANR-B and MviPBANR-C encoded 474 and 410 amino acids, respectively (Figure 1). Structurally, both MviPBANR-B and MviPBANR-C have 7-transmembrane (7-TM) regions and GPCR family 1 signature (Figure 1). Several motifs and residues involved in cell-surface localization, structural stabilization, and G protein coupling, as well as phosphorylation of PBANR by ligand binding and internalization of PBANR, were also identified (Figure 1).

MviPBANR isoforms, compared to those of other lepidopterans, in *M. vitrata* formed a clade with *Ostrinia nubilalis* in the Crambidae family (Figure 2). Phylogenetic analysis revealed that PBANRs of each species formed a clade according to the species, regardless of the isoforms.

### 3.2. Developmental Stage and Tissue-Specific Expression of PBANR Isoforms in M. vitrata

A qRT-PCR was performed to determine the in vivo expression patterns of MviPBANR-B and -C. Both isoforms were expressed at all developmental stages (Figure 3; Appendix A). In particular, the expression level of MviPBANR-C was four times higher than that of MviPBANR-B in adult females (Figure 3A).

Expression of MviPBANR isoforms was also examined in adult tissues. Both MviPBANR-B and -C were expressed in all tested tissues (Figure 3B,C). MviPBANR-C showed higher expression than that of MviPBANR-B in the female pheromone glands and male hair-pencils among other tested tissues (Figure 3B,C).

### 3.3. In Vitro Expression of PBANR Isoforms

For in vitro expression of MviPBANR isoforms, the full-length of MviPBANR-B and MviPBANR-C were transferred into pcDNA™3.1/Zeo vector, resulting in the expression vectors, pcMviPBANR-B and -C. Subsequently, pcMviPBANR-B and -C were transfected into HeLa cell lines, and selected with Zeocin. Successful transfection and selection of pcMviPBANR-B and -C were confirmed using PCR (Figure 4A). To investigate the response of MviPBANR isoforms to MviPBAN, transfected cells with pcMviPBANR-B and -C were treated with artificially synthesized MviPBAN at different concentrations, and the Ca^2+^ influx was observed using a confocal microscope. In pcMviPBANR-B transfected cells, Ca^2+^ influx was observed only in the positive control (ionomycin treatment) and no Ca^2+^ influx was observed in cells treated with MviPBAN (Figure 4B; Appendix A). In pcMviPBANR-C transfected cells, Ca^2+^ influx was induced when the cells were treated with more than 5 µM of MviPBAN and ionomycin (Figure 4B,C; Appendix A).

### 3.4. Change of Pheromone Production and Mating Behavior by Suppression of MviPBANR-C in M. vitrata

The suppressive effect of MviPBANR-C on sex pheromone biosynthesis and mating behavior was evaluated. Before MviPBANR-C suppression of RNAi, the diurnal expression of MviPBANR-C in the pheromone glands was measured every 4 h, starting from 2 h before photophase. MviPBANR-C was expressed throughout the photophase in the pheromone glands, and the expression level was relatively high in the scotophase. MviPBANR-C showed the highest expression level at 5 p.m. (Figure 5A; Appendix A); thus, RNAi suppression of MviPBANR-C was performed at 5 p.m.

MviPBANR-C suppression was observed on the first day after the injection of MviPBANR-C-specific dsRNAs, and its effects persisted for three days after the injection (Figure 5B; Appendix A). To examine the change in sex pheromone production after MviPBANR-C suppression, the amount of E10E12-16:Ald (the main component of the sex pheromone of *M. vitrata*) was measured using GC on the second day after the injection. During MviPBANR-C suppression, the amount of E10E12-16:Ald decreased by two-thirds compared to that of the controls (Figure 5C). This result indicates that MviPBANR-C is directly involved in sex pheromone biosynthesis.

Finally, changes in mating behavior after MviPBANR-C suppression were investigated. On the first day after eclosion, no mating behavior was observed in both the controls or the individuals that received the MviPBANR-C suppression treatment with RNAi (Figure 5D). On the second day after eclosion, the mating rate was ≤20% in the individuals that received the MviPBANR-C suppression treatment (Figure 5D). On the third day after eclosion, all individuals in the control groups showed mating behavior, but the mating rate of MviPBANR-C-suppressed individuals was ≤20%, similar to the second day after eclosion (Figure 5D). These results show that MviPBANR-C suppression reduces sex pheromone production, and thus, reduces mating behavior.

## 4. Discussion

The process of sex pheromone biosynthesis, which induces mating behavior in female moths, is regulated by PBAN according to diurnal rhythms [31]. Therefore, it is necessary to study the interactions between PBANR and PBAN to understand the signal transduction process of sex pheromone biosynthesis in moths. In this study, we identified two PBANR isoforms, MviPBANR-B and MviPBANR-C, in the pheromone glands of *M. vitrata* (Figure 1). Structurally, the two MviPBANR isoforms harbor a typical feature of GPCRs. The third extracellular loop (ECL) between TM6 and TM7 plays a role in peptide ligand recognition [12]. In MviPBANR isoforms, several motifs have been predicted. Five N-glycosylation sites were identified from the N-terminal region of the MviPBANR isoforms (Figure 1). N-glycosylation occurring at the N-terminal region of PBANR is known to affect the stability of PBANR [32]. In *H. zea*, mutations in the N-glycosylation sites of PBANR inhibited PBAN-induced Ca^2+^ influx, thereby negatively affecting PBANR function [12]. However, the loss of N-glycosylation sites in PBANR did not affect ligand-induced internalization in *B. mori* [33]. In *H. zea*, both secondary messengers, cAMP and Ca^2+^, are used in signal transduction of sex pheromone biosynthesis whereas cAMP does not act in *B. mori* [13,34]. Furthermore, depending on these secondary messengers, it is unknown whether the N-glycosylation sites directly affect the interaction between PBAN and PBANR. In addition, four N-myristoylation sites were identified in the MviPBANR isoforms (Figure 1). N-myristoylation sites in GPCRs are important in the translational process and are related to protein anchoring on the membrane and cell signal transduction regulation [35,36].

PBANR shows different expression patterns in developmental stages and tissues depending on the species. In this study, MviPBANR-B and MviPBANR-C were expressed in all developmental stages and tissues of *M. vitrata*, regardless of sex (Figure 3). The same expression pattern as MviPBANR isoforms was also observed in other Crambidae, such as *O. nubilalis* and *O. furnacalis*. In some cases, species may belong to the same genus but have different numbers of PBANR isoforms, such as in the genus *Ostrinia*, in which *O. furnacalis* has one PBANR isoform, and *O. nubilalis* has three isoforms [37,38]. OnuPBANR-B and OnuPBANR-C in *O. nubilalis* were highly expressed in the pheromone glands of one-day-old females [38]. Three MbrPBANR isoforms (MbrPBANR-A, -B, and -C) were identified in the pheromone glands of *Mamestra brassicae* (family: Noctuidae) and expressed in all developmental stages and tissues. Among them, the expression level of MbrPBANR-C was the highest in the pheromone glands of two-day-old females [20]. The expression patterns of the MviPBANR isoforms in *M. vitrata* were consistent with that of the MbrPBANR isoforms. MviPBANR-C showed higher expression levels than that of MviPBANR-B in the pheromone glands of two-day-old virgin females (Figure 3C). These results suggest that the expression patterns of PBANR are affected by independent factors within species, regardless of the genus.

In addition, PBANR of *H. armigera* and *Pseudaletia separata* was specifically expressed in pheromone glands, unlike MbrPBANR, although they both belong to Noctuidae [19]. As reported in HarPBANR expression of *H. armigera* [39], MviPBANR isoforms were also expressed in hair-pencils in males corresponding to pheromone glands in females (Figure 3B). HarPBANR suppression by RNAi reduced the release of hair-pencil secretions [39], and PBAN plays in regulating hair pencil pheromones [9]. MviPBANR isoforms were also highly expressed in 5th instar larvae (Figure 3A). SliPBANR identified in *S. littoralis* larvae activates mitogen-activated protein kinase (MAPK) and Ca^2+^ influx in response to SliPBAN [40]. In *B. mori*, MAPK is known to be involved in prothoracicotropic hormone-induced ecdysone biosynthesis [41,42,43]. Based on these results, it is possible that the interaction of MviPBAN and MviPBANR is involved in the molting and pupation of *M. vitrata*.

PBANR is a GPCR and Ca^2+^ plays an important role in the cellular signaling cascade caused by ligand-GPCR interactions [44,45]. Several studies have reported that PBANR activation by PBAN induces Ca^2+^ influx via heterologous expression [11,12,13]. Ca^2+^ influx in the PBANR-expressed cells induced by PBAN is quantitatively different among PBANR isoforms. Ca^2+^ influx induced by HarPBAN was observed only in sf9 cells expressing HarPBANR-B and HarPBANR-C in *H. armigera* [19]. In *O. nubilalis*, Ca^2+^ influx was observed only in sf9 cells expressing OnuPBANR-C in response to OnuPBAN [38]. In the present study, when HeLa cells expressing the MviPBANR isoforms reacted with MviPBAN, Ca^2+^ influx was observed only in cells expressing MviPBANR-C (Figure 4).

RNAi has been widely used as a tool to study gene function in organisms, including insects [46]. Functional analyses of PBANR in moths have mainly demonstrated the reactivity of PBANR to PBAN. However, in a few species such as *P. xylostella* and *S. frugiperda*, the function of PBANR was analyzed after PBANR suppression by RNAi in the pheromone glands of female adults [16,47]. In this study, prior to RNAi treatment, we measured the diurnal expression of MviPBANR-C in response to MviPBAN in the pheromone glands of *M. vitrata*. The expression level of MviPBANR-C was the highest at 2 h before the scotophase (5 p.m.) (Figure 5), which coincides with MvipgFAR expression [26]. RNAi suppression of MviPBANR-C was performed 2 h before the scotophase (5 p.m.), and this suppression reduced sex pheromone production as well as mating behavior (Figure 5). These results demonstrate that MviPBANR-C reacts with MviPBAN and is involved in pheromone biosynthesis and mating behavior. Here, we focused on RNAi regions on PBANR sequences in three moth species. In the case of *P. xylostella* and *S. frugiperda*, RNAi was performed on the TM region containing ECL, which is important for ligand binding [16,47]. However, RNAi suppression was performed on the C-terminal tail of MviPBANR-C, which is different from that of MviPBANR-B. Except for an N-myristoylation site, the C-terminal tail region used for RNAi suppression of MviPBANR-C does not contain an important motif for ligand binding, suggesting that N-myristoylation suppression may negatively affect cell signal transduction (Figure 5) [35,36].

## 5. Conclusions

In this study, we performed a functional analysis of MviPBANR isoforms involved in signal transduction of pheromone biosynthesis in *M. vitrata*, and only two MviPBANR isoforms were identified from the pheromone glands. Among these isoforms, only MviPBANR-C induced Ca^2+^ influx by responding to MviPBAN. MviPBANR-C is involved in sex pheromone biosynthesis in *M. vitrata* by interacting with MviPBAN, which affects sex pheromone production and mating behavior. These results suggest that MviPBANR-C is a potential pest control factor.

## Figures and Tables

**Figure 1 cells-12-01410-f001:**
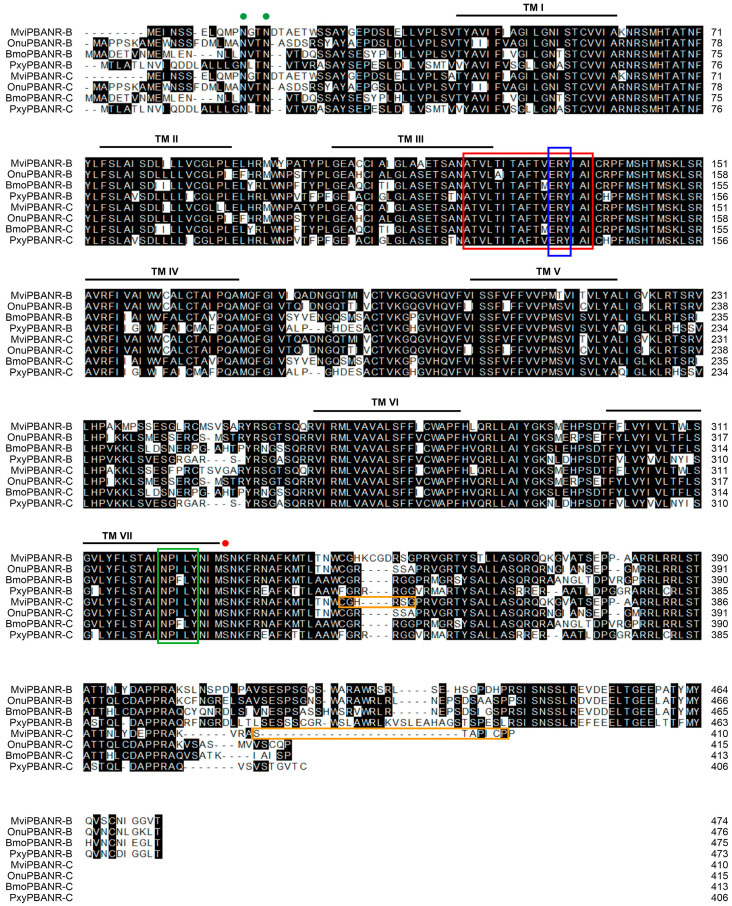
Amino acid sequence alignment of MviPBANR-B and MviPBANR-C with other lepidopteran PBANR isoforms. The alignment was performed using Crystal W. MviPBANR isoforms containing a 7-transmembranes (TM; black line) region, GPCR family 1 signature (red box). Several motifs and residues involved in cell-surface localization (green spots), structural stabilization and G protein coupling (blue box), phosphorylation of PBANR by ligand binding (red spot), and internalization of PBANR (green box) were identified. Primer regions for MviPBANR-C-specific dsRNA synthesis were indicated with orange boxes. N-glycosylation sites (^4^N–^7^E, ^12^N–^15^N, ^15^N–^18^A, ^52^N–^55^T, ^62^N–^65^M, ^120^N–^123^V) and N-myristoylation sites (^13^G–^18^A, ^48^G–^53^I, ^51^G–^56^C, ^369^G–^374^E) were identified. The abbreviated species names are as follows: Bmo, *Bombyx mori* (-B, AEX15643; -C, AEX15640); Mvi, *Maruca vitrata* (this study); Onu, *Ostrinia nubilalis* (-B, AGL12067; -C, AGL12068); Pxy, *Plutella xylostella* (-B, JAV44789; -C, JAV44788).

**Figure 2 cells-12-01410-f002:**
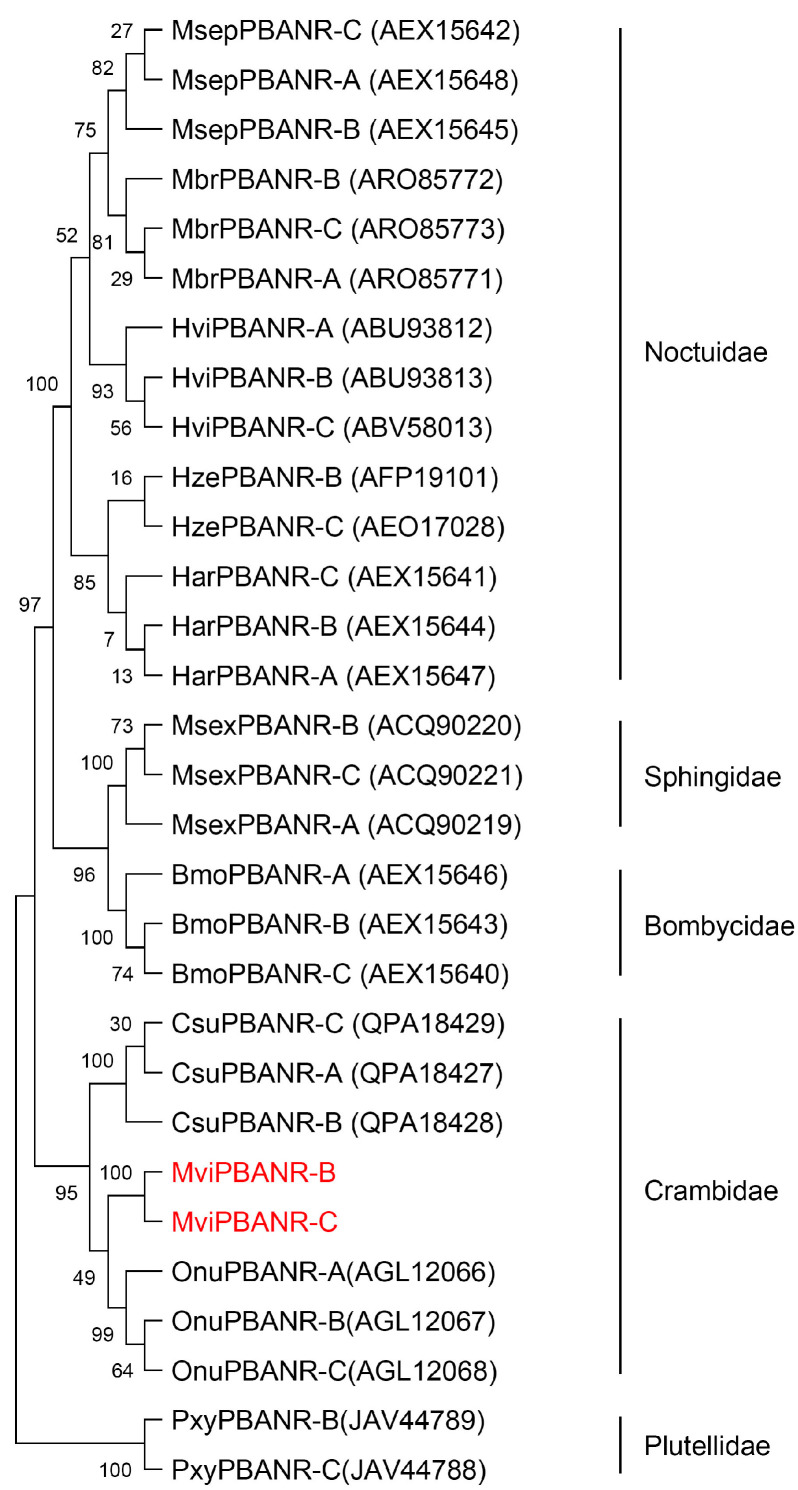
Phylogenetic analysis of MviPBANR-B and MviPBANR-C with other lepidopteran PBANR isoforms. A phylogenetic tree was constructed with the maximum likelihood method using multiple alignment of amino acid sequences. Numbers above branches are bootstrap values with 1000 repetitions. Bars indicate the number of estimated amino acid changes per 100 amino acids. The abbreviated species names correspond to: Bmo, *Bombyx mori*; Csu, *Chilo suppressalis*; Har, *Helicoverpa armigera*; Hvi, *Heliothis virescens*; Hze, *Helicoverpa zea*; Mbr, *Mamestra brassicae*; Msep, *Mythimna separata*; Msex, *Manduca sexta*; Mvi, *Maruca vitrata*; Onu, *Ostrinia nubilalis*; Pxy, *Plutella xylostella*.

**Figure 3 cells-12-01410-f003:**
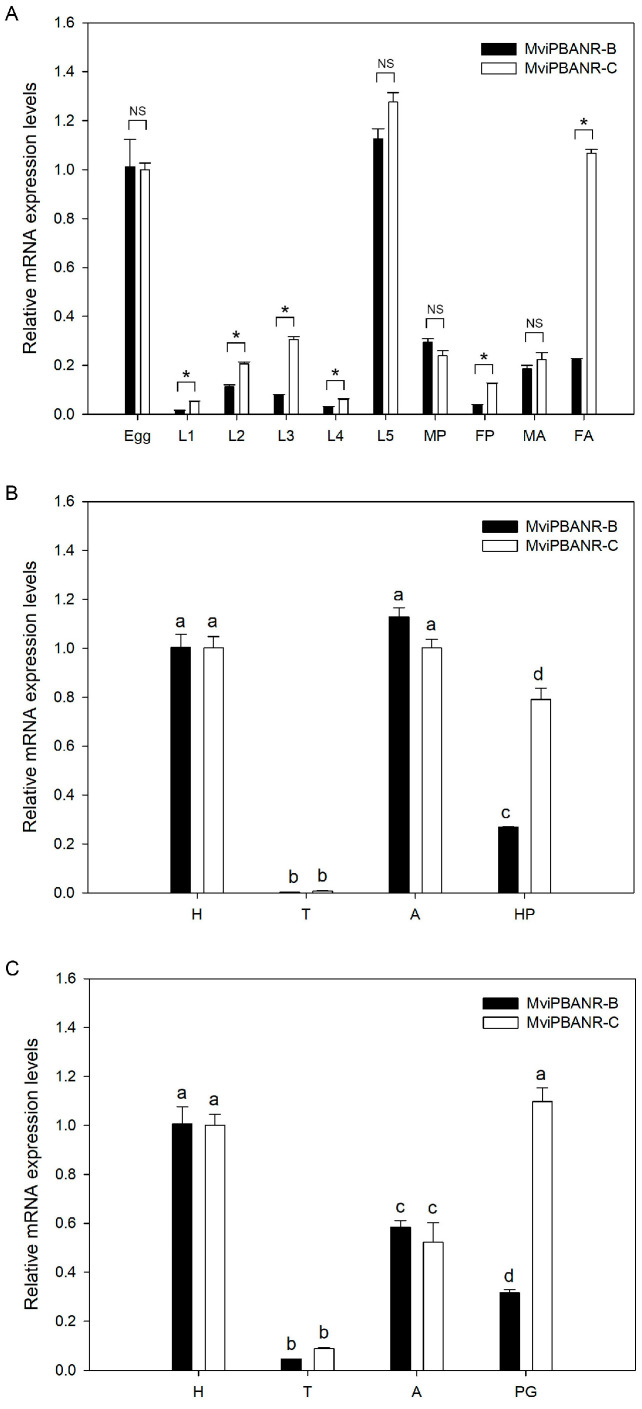
In vivo gene expression of MviPBANR-B and MviPBANR-C. (**A**) Developmental stage-specific expression. Egg, different instar larvae (L1–L5), pupa (P), and adult (A) were analyzed. For pupae and adults, male (M) and female (F) individuals were analyzed. The statistical differences were analyzed by *t*-test (* *p* < 0.001; NS, no significance). (**B**) Tissue-specific expression in male adults; head (H), thorax (T), abdomen (A), and hair-pencil (HP) were analyzed. The statistical differences were analyzed by ANOVA (*p* < 0.001). (**C**) Tissue-specific expression in female adults; head (H), thorax (T), abdomen (A), and pheromone gland (PG) were analyzed. ANOVA was used to analyze statistically significant differences (*p* < 0.001). EF1-α was used as a positive control for the qRT-PCR.

**Figure 4 cells-12-01410-f004:**
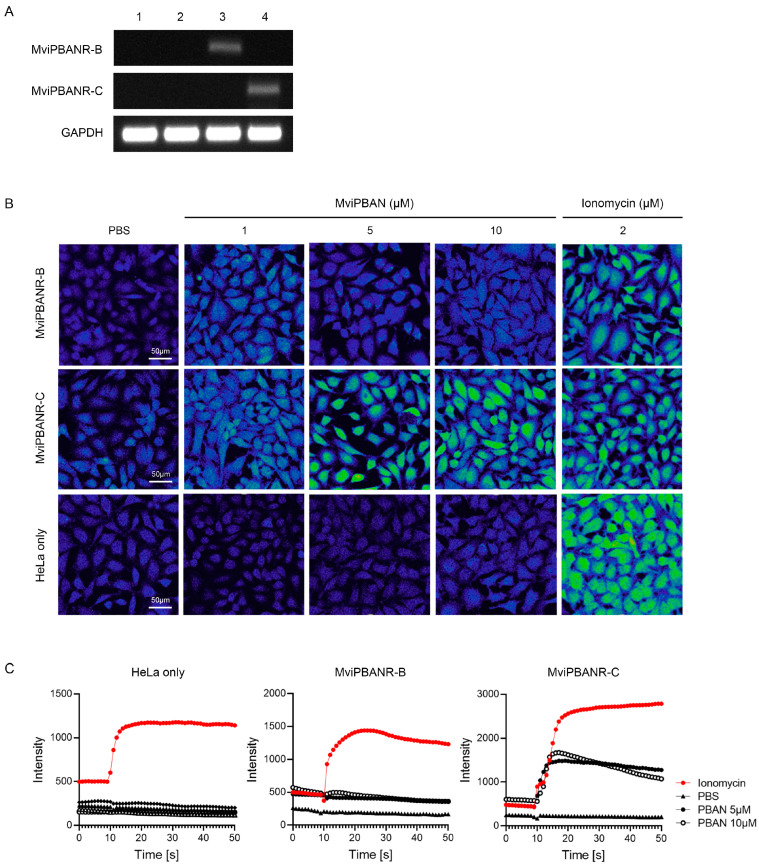
Functional analysis of heterologous expression of MviPBANR isoforms in HeLa cells. (**A**) Transfection of MviPBANR isoforms was confirmed using PCR. Glyceraldehyde 3-phosphate dehydrogenase (GAPDH) was used as a positive control. 1, non-transfected HeLa cells; 2, HeLa cells transfected with PBS; 3, HeLa cells transfected with MviPBANR-B; 4, HeLa cells transfected with MviPBANR-C. (**B**) Reactivity of MviPBANR isoforms to MviPBAN expressed in HeLa cells was confirmed by Ca^2+^ influx. PBS was used as a negative control and ionomycin was used as a positive control. (**C**) Signal transduction of MviPBANR by MviPBAN over time was confirmed. PBS was used as a negative control, and ionomycin was used as a positive control.

**Figure 5 cells-12-01410-f005:**
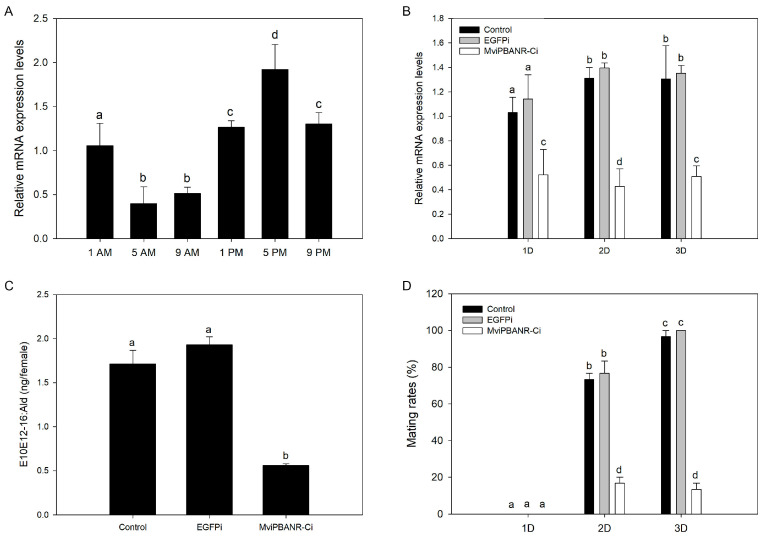
Pheromone production and mating behavior after MviPBANR-C suppression by RNAi in female adults of *M. vitrata*. (**A**) Diurnal expression of MviPBANR-C in pheromone glands. The expression levels of MviPBANR-C against EF1-α were measured by qRT-PCR. (**B**) Suppression of MviPBANR-C after RNAi in the pheromone glands. The expression levels of MviPBANR-C against EF1-α measured by qRT-PCR between control and RNAi group. EGPF-specific dsRNA was injected as a negative control. (**C**) Pheromone production after MviPBANR-C suppression in the pheromone glands. The amount of E10E12-16:Ald, the major sex pheromone component of *M. vitrata*, was measured after MviPBAN injection using GC analysis. EGPF was suppressed as a negative control. (**D**) Reduction in mating rates after MviPBANR-C suppression in adults. EGPF was suppressed and used as a negative control. Different letters above each bar indicate a statistical difference by ANOVA (*p* < 0.001).

## Data Availability

Not applicable.

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
