# Peer review of "Functional Analysis of Pheromone Biosynthesis Activating Neuropeptide Receptor Isoforms in Maruca vitrata"

_cells, 2023, doi:10.3390/cells12101410_

Round 1

Reviewer 1 Report

The authors performed a series of work to study on the Function of pheromone biosynthesis activating neuropeptide in Maruca vitrata. The results are interesting, and most of the method and data analysis are solid. However, there are still some problems need to be solved.

The most problem I care about is that in the line 116, "elongation factor 1-α(EF1-α)", and why the authors chose only one reference gene for normalizing data in the work of qRT-PCR. In my opiniom, at least two reference genes need to be used in qRT-PCR, so another reference gene needs to be added in the work.

Moreover, throughout the MS, there are various problems about scientific writing. For example, about the P value, the letter "P" should be italic. And, similar errors of scientific writing are too many to be list in the MS.

Reviewer 2 Report

This study explored the characterization of two PBAN receptors in a crambid moth.  The manuscript was well written and clearly showed that the isoform PBANR-C is involved in PBAN signal transduction.  This was done through heterologous expression and functional stimulation of calcium influx and the use of RNAi to knockdown the receptor in female moths. I just have a few specific comments as listed below.

Line 46 – Actually about 90% of Lepidoptera PBAN sequences an FSPRL-amide C-terminal ending. 

Line 111 – Indicate if the abdomen was analyzed without the pheromone glands or hair pencils.

Line 141 – The use of the word ‘stained’ is inappropriate.  I would just remove the words ‘and stained’ from the sentence.

Line 178 – check the sequence of the MviPBAN.  Should it be a FNPRL-amide ending.

Line 182 – the name of the GC is missing from the description.

Line 314 – change to: The process of sex pheromone biosynthesis,

Line 318 – indicate the B and C isoforms could be involved in sex pheromone signal transduction.  The study demonstrates that only the C isoform is apparently involved in signal transduction.

Line 328 – actually both B. mori and H. zea use calcium as a second messenger; they differ in that B. mori does not use cAMP.

Line 364 – PBANR is a GPCR. The PBANR is a GPCR and calcium plays…

Line 373 – The discussion of where the PBANR isoforms are found is confusing.  Heterologous expression using different cells could account for differences found in the various studies. The last point about PBANR-A being found in the cytoplasm is not entirely correct in that it was also found in the cell membrane but was apparently transferred to the cytoplasm faster in Sf9 cells.

In a few places the English needs some editing, otherwise a well written manuscript.

Round 2

Reviewer 1 Report

No more further questions.

Reviewer 3 Report

Authors have now adequately answered all the queries, and MS is now acceptable for publications.